# Psychometric Evaluation of the Korean Version of the Personhood in Dementia Questionnaire Using Rasch Analysis

**DOI:** 10.3390/ijerph16234834

**Published:** 2019-12-01

**Authors:** Da Eun Kim, Young Ko, Paulette V. Hunter, Ju Young Yoon

**Affiliations:** 1School of Nursing, University of Wisconsin-Madison, Madison, WI 53705, USA; dkim622@wisc.edu; 2College of Nursing, Gacheon University, Incheon 21936, Korea; youngko@gachon.ac.kr; 3Department of Psychology, St. Thomas More College, University of Saskatchewan, Saskatoon, SK S7N 0W6, Canada; phunter@stmcollege.ca; 4Research Institute of Nursing Science and College of Nursing, Seoul National University, Seoul 03080, Korea

**Keywords:** person-centered care, dementia, attitude, long-term care, psychometrics

## Abstract

There is an increasing awareness of the need to promote behaviors consistent with the understanding that individuals with dementia deserve adequate respect. Person-centered attitudes on the part of a care facility’s staff can affect care practices and relationships with residents. This study examined the psychometric properties of the Korean version of the Personhood in Dementia Questionnaire (KPDQ), which measures staff’s person-centered attitudes toward individuals with dementia. The KPDQ was translated and adapted based on commonly used guidelines from the World Health Organization. For psychometric testing, the data obtained from a total of 269 participants in 13 long-term care facilities were analyzed. Factor analysis, item fit, convergent validity, and known-group validity were examined. Reliability and differential item functioning (DIF) based on Rasch analysis were also assessed. The KPDQ consists of 20 items with three subscales (“agency”, “respect for personhood” and “psychosocial engagement”). Item fit statistics indicated that each item fits well with the underlying construct. The KPDQ demonstrated satisfactory convergent validity, known-group validity and internal consistency reliability. There was no DIF by subgroup according to age or educational status. Results indicated that the KPDQ is a reliable and valid tool for measuring long-term care staff’s beliefs about personhood.

## 1. Introduction

Increasing evidence indicates that person-centered care is an essential dimension of care for patients with dementia living in long-term care facilities [1]. Person-centered care supports health by attending to the individual patient’s values and preferences [2]. It focuses on improving quality of life by delivering individualized care based on the dignity of each human being [3]. Kitwood ([4], p. 8) suggested person-centered care as the basis for dementia care and developed the concept of personhood in dementia care, defining it as “a standing or status that is bestowed upon one human being, by others, in the context of relationship and social being. It implies recognition, respect, and trust.” Person-centered care in long-term care settings has a beneficial effect on residents’ psychological well-being and behavioral symptoms, in addition to reducing the need for psychotropic medication [5].

Person-centered care is multi-dimensional. The person-centered nursing framework of McCormack and McCance [6] divides person-centered care into four constructs: staff attributes, care environment, care processes, and expected outcomes (e.g., resident satisfaction with care, a feeling of well-being). The beliefs and attitudes of the staff are regarded as antecedents to providing person-centered care effectively [6,7]. Previous studies have found that, in daily practice, staff beliefs about personhood influence their caring behaviors toward older adults with dementia [8,9]. Staff attitudes toward residents are a major factor influencing the closeness of relationships between residents and staff. Such relationships have been emphasized as critical to person-centered nursing in improving residents’ quality of life [10,11]. Furthermore, the person-centered attitudes of long-term care staff toward residents with dementia are associated with good staff outcomes, including higher job satisfaction and less stress [12,13]. As the values and beliefs shared among staff can have a significant impact on organizational culture, healthy staff perspectives on residents with dementia should be considered vital.

Assessment of person-centeredness is the first step when seeking to apply person-centered care and improve care quality [14]. Although personhood perception is a prerequisite for effective person-centered care, no instrument has yet been developed in Korea to assess staff attitudes toward person-centered care. Two instruments are currently available to assess the attitudes of long-term care staff toward personhood: the Tool for Understanding Residents’ Needs as Individual Persons (TURNIP) [15] and the Personhood in Dementia Questionnaire (PDQ) [8]. TURNIP is an intervention tool developed in Australia that comprehensively explores five dimensions of person-centered care, including care environment, staff knowledge of dementia, organization, care practice, and staff attitudes toward persons with dementia [15]. However, TURNIP includes a subscale related to the attitudes toward persons with dementia so researchers do not separately utilize the attitude domain only. Furthermore, the reliability and validity of the total TURNIP were satisfactory; however, the Cronbach’s alpha coefficient for the attitudes subscale was 0.63, which is slightly lower than the acceptable criterion (≥0.7) [15]. Originally developed in Canada, the PDQ is an instrument that can be used to assess the person-centered attitudes of long-term care staff [8]. To generate the original PDQ items, the authors interviewed staff members about their experiences in dementia care to reflect a variety of perspectives. They then analyzed the responses for themes related to personhood, drafted an initial pool of questionnaire items based on these themes, and refined the questionnaire items via psychometric testing [8]. The PDQ has demonstrated satisfactory internal consistency reliability, resistance to social desirability bias, and convergent and discriminant validity [8]. The PDQ has been employed in several subsequent studies to examine the association between beliefs about personhood and person-centered dementia care [9], and to identify the relationship between resilience and perceptions of resident personhood among long-term care staff [16].

As beliefs and attitudes can vary depending on the cultural context, a rigorous process was followed to culturally adapt the PDQ to assess attitudes toward residents’ personhood among staff in Korean long-term care facilities. The objective of this study was to translate the PDQ from English to Korean, adapt it culturally, and evaluate the reliability and validity of the Korean version using the Rasch model.

## 2. Materials and Methods

### 2.1. Study Design

This study was based on the methodological research design to examine the psychometric properties of a Korean version of the PDQ (KPDQ).

### 2.2. Transcultural Translation and Adaptation of the Instrument

After the original author gave us written approval for the development of the KPDQ, we adapted the instrument using the translation/adaptation guidelines of the World Health Organization [17]. First, a nursing graduate student who had lived in the USA for more than 10 years translated the instrument from English into Korean (forward translation). Second, three bilingual research experts with considerable experience in person-centered care assessed whether the Korean version was conceptually similar to the original instrument, and whether it was culturally and institutionally appropriate (expert panel). They modified several items. Third, the revised Korean version was back-translated into English by another independent translator with a Nursing Bachelor degree living in the USA who is bilingual in English and Korean (back-translation). Fourth, the expert panel again assessed the compatibility of the original and back-translated versions, and confirmed that no significant differences in terms of either content or context were apparent. Then, cognitive interviews were conducted with five long-term care staff to confirm the clarity of all items and applicability of the instrument (cognitive interviewing). Finally, after careful proofreading, minor modifications were made with reference to the culture, language, and long-term care contexts of Korea.

### 2.3. Participants

This psychometric testing study utilized part of the data from the parent study, which examined the overall relationships among person-centered care and care outcomes in Korean long-term care facilities from the perspectives of residents and staff. Data were collected from November 2016 to February 2017. The inclusion criterion for participation in this study was the staff who had work experience of one month or more in a long-term care facility providing care to persons with dementia. The parent study was conducted under the approval of the Institutional Review Board (IRB) and another IRB approval of exemption was obtained for the present study. We guaranteed anonymity and confidentiality, and obtained written informed consent prior to data collection. For this study of the psychometric properties of the KPDQ, we used 269 participants (nursing staff, personal care workers, social workers, and physical therapists) in 13 Korean long-term care facilities. To achieve stable factor analysis results, the sample size must be at least 10 times the number of items (in this case, 20 items) [18]. Since, according to this standard, the number of study participants should have been more than 200, the actual number of participants in this study (*n* = 269) was sufficient [18].

### 2.4. Instruments

#### 2.4.1. Personhood in Dementia Questionnaire (PDQ)

The PDQ was originally developed in English by Hunter et al. [8] to measure beliefs about personhood. The questionnaire consists of 20 items each scored on a seven-point Likert scale (1 = disagree strongly to 7 = agree strongly). The total scores range from 20 to 140; higher scores indicate that the attitude toward personhood is more person-centered. In the original study, the PDQ exhibited satisfactory convergent and discriminant validity, based on correlations with the personhood subscale of the Person-Directed Care instrument (r = 0.52, *p* < 0.01) and the physician attitudes subscale of the Cancer Attitudes Questionnaire (r = −0.24, *p* = 0.11) [8]. The internal consistency reliability was also satisfactory (Cronbach’s alpha = 0.79) [8].

#### 2.4.2. Person-Centered Climate

Given that the sub-concepts of person-centered care (i.e., person-centered attributes of caregivers, care environment, and care practice) are correlated [6], person-centered climate was measured using the Korean version of the Person-centered Climate Questionnaire-Staff (KPCQ-S) to assess convergent validity [19]. The KPCQ-S is a 17-item questionnaire that measures the person-centered characteristics of health care environments as perceived by staff members [19]. Each item is rated on a six-point Likert scale from one to six. In a validation study of the KPCQ-S at Korean long-term care facilities, the reliability (Cronbach’s alpha) was 0.91 [19].

#### 2.4.3. Job Satisfaction

Job satisfaction was measured using the job satisfaction subscale of the Korean version of the Copenhagen Psychological Work Environmental Measure (COPSOQ-K) [20] to examine known-group validity. This was appropriate since job satisfaction is positively related to staff members’ attitudes toward residents with dementia [13]. The job satisfaction subscale consists of four items, each rated on a four-point Likert scale [20]. The mean scores range from one to four, with higher scores indicating higher job satisfaction. Cronbach’s alpha was 0.78 in the validation study of the Korean version [20] and 0.85 in the present study.

#### 2.4.4. Demographics

Personal demographic characteristics including age, gender, educational attainment and marital status were assessed. Work-related demographic characteristics including occupation, work experiences, work shift (rotating shift or fixed shift), and monthly income were examined.

### 2.5. Statistical Analyses

To examine the construct validity of the KPDQ, exploratory factor analysis (EFA) was conducted, using robust weighted least-squares estimation as the factor extraction method and geomin rotation as the factor rotation method. Confirmatory factor analysis (CFA) was then tested to confirm the factor structure determined by EFA. Model fit indices included the chi-square test, the comparative fit index (CFI) and the Tucker-Lewis index (TLI) ≥ 0.90, and the standardized root mean-square residual (SRMR) ≤ 0.08 [21].

Based on item response theory (IRT), item fit was assessed to identify how well each item contributed to measuring a single latent trait; i.e., the strength of the staff’s beliefs about the personhood of long-term care residents who have dementia [22]. Item fit was assessed using information-weighted fit (INFIT) and outlier-sensitive fit (OUTFIT) mean square (MnSq). An MnSq fit statistic of 0.5–1.5 is considered adequate [23,24]. This range was used as the criterion for determining the deletion of items. INFIT or OUTFIT values lower than the criterion may indicate redundancy, implying that an item is not uniquely useful. Values higher than the criterion may indicate that an item is not relevant to the instrument [25].

Convergent validity was tested by performing correlation analysis between the KPDQ and person-centered climate using the KPCQ-S. The criterion for acceptable convergent validity was ≥ 0.4 [26]. In terms of known-group validity, the differences in KPDQ scores according to job satisfaction were analyzed using the independent *t*-test. We dichotomized staff responses at the mid-point of the scale. Staff who responded positively (>2.5 points) were classified as exhibiting a high level of job satisfaction, and those who responded negatively (≤2.5 points) were classified as having low-level job satisfaction.

Internal consistency reliability was evaluated for each sub-domain using Cronbach’s alpha coefficient and McDonald’s coefficient omega (Ω) [27]. The reliability cut-off point was set to ≥ 0.7 [28]. In addition, the person separation index (PSI), a Rasch-based person-separation reliability statistic, was calculated. This statistic indicates how well an instrument classifies participants into groups with reference to the scores [29]. As the Rasch model investigates the reliability of items and individuals, Rasch-based reliability statistics have recently become preferred to Cronbach’s alpha [30]. A high PSI (≥2.0) indicates that individuals estimated to score high really do score higher than individuals who score lower [31].

Differential item functioning (DIF) based on IRT was assessed to identify whether respondents from different groups with the same level of the latent trait showed different item response functions. In this study, DIF between subgroups according to age (≤52 years old vs 53 years old or more, based on a median age of 52) or educational attainment (high school or less vs college diploma or more) was assessed. More than a 0.5 logit difference indicates the DIF effect has a meaningful impact on measures [32].

Descriptive statistics, correlation and independent *t*-tests were performed using IBM SPSS statistics ver. 24.0 (IBM Corp., Armonk, NY, USA). Factor analysis and calculating omega were performed using Mplus ver. 8.3 (Muthén & Muthén, Los Angeles, CA, USA). All Rasch analyses were performed using Winstep ver. 4.4.5 (Winsteps, Beaverton, OR, USA).

## 3. Results

### 3.1. Descriptive Characteristics of the Study Participants

The study participants were primarily female (95.5%) with a mean age of 49.61 years (Table 1). They included nursing staff (33.9%), personal care workers (46.5%), social workers (9.3%), and physical therapists (10.4%). The mean working duration was 48.16 months and about half were shift workers. The mean KPCQ-S score was 4.70 on a one to six Likert scale. The mean job satisfaction score was 2.89 on a one to four Likert scale.

### 3.2. Item Analysis

The mean KPDQ total score was 95.79 (Table 2). The mean scores of each item ranged from 3.51 to 6.10. The item with the highest mean score was a reverse-coded item, item 18 (“Residents with advanced dementia are no longer persons like you and me, because they do not think and reason logically.”). The item with the lowest mean score was item 2 (“Most residents with dementia are still capable of making some informed choices about their lives.”).

### 3.3. Validity

#### 3.3.1. Factor Analysis

First of all, the Kaise-Meyer-Olkin (KMO) test and Bartlett’s test of sphericity were conducted to assess the suitability of the data set for factor analysis. In this study, the KMO value was 0.84 and the Bartlett’s test of sphericity was significant (x^2^ = 1907.28, *p* < 0.001), which means that the sample was appropriate for factor analysis. The number of factors was explored based on a scree plot and Eigen values. The number of factors with an Eigen value of 1 or more was five (6.55, 2.27, 1.85, 1.21, and 1.12, respectively). Each factor had more than three items [33]. However, in four-factor and five-factor models, there were factors on which only one or two items were loaded. Therefore, a three-factor model was determined as the final factor structure of the KPDQ. In the EFA, all items except one were loaded (>0.40) in at least one factor (Table 2). The loading value of item 14 (0.31) was lower than the desirable criterion, but higher than the minimum recommended item factor loading value of 0.30 [34]. Also, as item 14 is related to the concept of the Factor 3, the research team decided to include item 14 under factor 3. Then CFA was followed to verify the three factor structure identified through EFA. Results indicated that the three-factor model was appropriate with an acceptable fit (CFI = 0.920, TLI = 0.909, and SRMR = 0.069). Each of the KPDQ subscales measures the strength of the respondent’s beliefs about one aspect of the personhood of long-term care residents who have dementia. Meaning of the three factors are as follows:Factor 1:agency (beliefs about residents’ capacity for self-determination)Factor 2:respect for personhood (beliefs about residents’ personhood and moral status)Factor 3:psychosocial engagement (beliefs about residents’ capacity for psychological and social engagement)

#### 3.3.2. Item Fit

Item fit statistics were assessed to examine item concurrence with overall long-term staff attitudes about the personhood of residents who have dementia. Two fit indices, the INFIT and OUTFIT MnSq statistics for each item, are listed in Table 2. INFIT MnSq statistics ranged from 0.64 to 1.44 and OUTFIT MnSq statistics ranged from 0.63 to 1.34. All KPDQ items showed satisfactory fit of observed data to Rasch model-expected data.

#### 3.3.3. Convergent Validity

Convergent validity was tested by analyzing the correlations of the KPDQ and the KPCQ-S using Pearson’s correlation coefficient. The KPDQ was significantly correlated with the KPCQ-S (r = 0.40, *p* < 0.001). Convergent validity was satisfactory.

#### 3.3.4. Known-Group Validity

A known-group comparison using an independent *t*-test between the two groups according to job satisfaction scores was conducted. The mean KPDQ total score of the high job satisfaction group was 96.96, whereas that of the low job satisfaction group was 89.10 (Table 3). The high job satisfaction group reported significantly greater person-centered attitudes than the low job satisfaction group (t = −3.04, *p* = 0.003).

### 3.4. Reliability

As shown in Table 2, the Cronbach’s alpha coefficient of the whole KPDQ scale was 0.86. The corrected item total correlations ranged from 0.33 to 0.62. The Cronbach’s alpha values when items were individually deleted ranged from 0.85 to 0.86. Cronbach’s alpha coefficient was 0.76 for agency, 0.77 for respect for personhood, and 0.82 for psychosocial engagement. Coefficient omega was 0.80 for agency, 0.83 for respect for personhood, and 0.86 for psychosocial engagement. The PSI based on the Rasch model was 2.35, which corresponds to the criterion (≥2.0). Overall, the results showed the internal consistency reliability of the KPDQ was satisfactory.

### 3.5. Differential Item Functioning

We assessed DIF across the subgroups of age and educational attainment to assess the possibility of differences in responses to the KPDQ items. There was no DIF (contrast of >0.5 logits) by age or educational attainment.

## 4. Discussion

To develop the KPDQ, we adapted the English version of the PDQ and conducted Rasch analysis to examine internal consistency reliability, item fit and DIF. Factor analysis, convergent validity and known-group validity were also tested. The results showed that the instrument reliably and validly measured the person-centered attitudes of staff working in Korean long-term care facilities.

Factor analysis demonstrated the appropriateness of a three-factor structure for the KPDQ in Korean long-term care facilities (i.e., agency, respect for personhood, and psychosocial engagement). The structure was identified for the first time in this study because factor analysis was not conducted in the original PDQ development study in Canada [8]. Subscales were labeled to reflect the theory guiding the original PDQ item pool [8] through discussion with the PDQ’s original researcher. Agency is defined as being able to act in accordance with motivation, desire and will [35]. The ‘agency’ subscale includes items that measure the strength of the respondent’s attitudes regarding whether people with dementia have the ability and the right to make decisions. The ‘respect for personhood’ subscale includes items measuring the strength of the respondent’s beliefs about the extent to which residents deserve respect. The ‘psychosocial engagement’ subscale includes items related to the suggestion by Post [36] that personhood should be linked more strongly to affect and the capacity for relationships.

Item fit analysis for each item was performed to insure that all items matched expected responses based on the Rasch model. INFIT and OUTFIT MnSq statistics for each item were within the acceptable range, indicating a satisfactory fit to the global underlying trait; i.e., the respondent’s beliefs about one aspect of the personhood of long-term care residents who have dementia. These results indicate that the KPDQ had no redundant items or poorly constructed items.

In terms of convergent validity, the KPDQ was significantly correlated with the KPCQ-S, consistent with the correlation between the PDQ and the modified personhood subscale of the Person Direct Care (PDC) found in a previous study (r = 0.52, *p* < 0.01) [8]. In this study, known-group comparison was conducted to examine whether the KPDQ scores were able to discriminate between two groups according to job satisfaction, which is known to differ depending on person-centered attitudes [13]. The independent *t*-test showed that the mean KPDQ score was significantly higher in the high job satisfaction group than in the low job satisfaction group. This is consistent with the results of previous researches showing that the job satisfaction of long-term care staff is positively associated with a person-centered attitude [12,13]. Through these analyses, convergent validity and known group validity were confirmed to be satisfactory.

The Cronbach’s alpha coefficient of the entire scale (0.86) indicated acceptable internal consistency, indeed slightly higher than that of the original version (0.79–0.81) [8]. The corrected item total correlations (0.33–0.62), together with the Cronbach’s alpha values when items were individually deleted (0.85–0.86), indicated that the reliability of the KPDQ based on the degree of within-scale item intercorrelation was satisfactory. Coefficient omega is recommended as a sensible index of internal consistency, as it has less possibility for overestimation or underestimation of reliability [37]. In this study, the value of coefficient omega for the three subscales was above 0.70. The evidence indicates the KPDQ is sufficiently reliable. Moreover, the PSI (2.35) indicated good internal consistency. That is, participants with higher KPDQ scores can be estimated to actually have higher person-centered attitudes than those with lower KPDQ scores. Therefore, the KPDQ exhibited satisfactory reliability.

The mean KPDQ total score (95.79) was lower than the mean score in the previous study measured in Canada with the original PDQ instrument (116.80) [8]. Presumably, this is due to between-country differences. Korea has aged rapidly in recent years. The number of long-term care facilities has increased to meet the needs of older adults and their family members. The qualitative aspects of long-term care (e.g., resident and family satisfaction) have received little attention. Residents’ perceived needs are not matched by staff services; person-centeredness has not become a core care principle [38]. However, as the elderly with physical and psychological impairments can live for many years in long-term care facilities, person-centered care placing priority on residents rather than caregivers is critical. Staff require dedicated education and training on both philosophical topics (e.g., what is person-centered care? what are basic human rights?) and practical aspects of person-centeredness (e.g., person-centered communication, a person-centered approach to behavioral symptoms) [9].

In this study, the highest mean scores were seen for items 18 (“Residents with advanced dementia are no longer persons like you and me, because they do not think and reason logically.”) and 7 (“All residents with dementia should be treated with respect.”). These results were similar to those of the previous study in Canada [8]. However, the mean score for a reverse-coded item (item 16; “Residents with dementia who whine a lot should be isolated.”), which had a high mean score in the previous study [8], was found to be below average in this study. This implies that the Korean long-term care staff who participated in the present study tended to think that they should isolate such residents. As human beings are fundamentally social beings, social relationships and interactions with others could help people with dementia to maintain personhood [39]. Therefore, isolation should be avoided when possible in providing person-centered care.

In this study, the item with the lowest mean score was item 2 (“Most residents with dementia are still capable of making some informed choices about their lives.”). This indicates that the staff members in Korean long-term care facilities have little awareness of the importance of decision-making and patient engagement among elderly people with dementia, though this is fundamental to person-centered care. However, each resident’s choice should be central to care delivery because each long-term care resident has a different preference for daily care and activity [40]. Thus, continuous effort is required to improve staff attitudes and promote awareness about the importance of respecting the needs and values of persons with dementia and reflecting these unique values in care practice.

Our study had several limitations. First, we employed convenience sampling and thus cannot exclude selection bias. Second, given the cross-sectional nature of the study, we cannot comment on predictive validity or test-retest reliability. However, we have performed groundwork for evaluation of staff attitudes toward personhood and exploration of the relationships between these attitudes and other aspects of person-centered care (e.g., the environment, care practices, and resident outcomes).

## 5. Conclusions

The KPDQ can be administered to Korean long-term care staff as an indicator of their person-centered attitudes toward persons with dementia. In 2000, the Institute of Medicine (IOM) [41] included person-centeredness as an aim for health care quality. In fact, it was suggested that person-centeredness should be considered a more important quality dimension than even safety or effectiveness [42]. Despite growing interest in person-centered care, staff attitudes have received little attention and have not been empirically examined in Korea. The KPDQ can be used to identify areas where there is a low awareness of person-centeredness among the staff. Based on this study, future research is needed to examine the relationships between attitudes and care practices, and to develop education programs that can enhance person-centered attitudes, with the ultimate aim of improving resident’s quality of life.

## Figures and Tables

**Table 1 ijerph-16-04834-t001:** Descriptive characteristics of study population (*N* = 269).

Variables	Categories	*n* (%)	Mean (SD)
*Personal factors*			
Age (years)			49.61 (11.28)
Female (ref: male)		257 (95.5)	
Education attainment	High school or less	117 (43.7)	
	College diploma	75 (28.0)	
	Bachelor degree or more	76 (28.4)	
With spouse (ref: no spouse)		216 (80.6)	
*Work-related factors*			
Type of occupation	Nursing staff	91 (33.9)	
	Personal care worker	125 (46.5)	
	Social worker	25 (9.3)	
	Physical therapist	28 (10.4)	
Working experiences (months)			48.16 (43.19)
Rotating shift (ref: fixed shift)		134 (50.0)	
Monthly income ($)			1677.32 (359.57)
	<1500	90 (34.5)	
	1500–1999	108 (41.4)	
	2000–2499	55 (21.1)	
	≥2500	8 (3.1)	
*Person-centered climate (KPCQ-S, 1–6)*			4.70 (0.57)
*Job satisfaction (1–4)*			2.89 (0.45)

SD: standard deviation; KPCQ-S: Korean version of the Person-centered Climate Questionnaire-Staff; Number of missing data: age = 1, education attainment = 1, marital status = 1, work experiences = 5, rotating shift = 1, monthly income = 8.

**Table 2 ijerph-16-04834-t002:** Means, internal consistency reliability, factor loadings and item fit statistics (*N* = 269).

Item Content	Mean (SD)	Item-Total Correlation	Cronbach’s α If Item Deleted	Factor Loadings			INFIT MnSq	OUTFIT MnSq
				F1	F2	F3		
Factor 1. Agency (Cronbach’s alpha = 0.76, Omega = 0.80)								
1. Residents with dementia have a sense of purpose.	4.05 (1.48)	0.36	0.86	0.78	0.01	0.03	1.05	1.07
2. Most residents with dementia are still capable of making some informed choices about their lives.	3.51 (1.49)	0.44	0.85	0.77	0.04	0.07	0.95	0.96
3. Residents with dementia have a basic right to make any choices they can about their care.	4.68 (1.42)	0.38	0.86	0.52	0.26	−0.01	0.99	0.99
6. Residents with dementia contribute to a sense of community within our long-term care facility.	4.51 (1.33)	0.44	0.85	0.54	0.30	−0.01	0.80	0.80
Factor 2. Respect for personhood (Cronbach’s alpha = 0.77, Omega = 0.83)								
4. Residents with very advanced dementia are so low-functioning that they are no longer persons. ^†^	5.81 (1.42)	0.48	0.85	0.01	0.74	0.12	1.42	1.29
5. Residents with end-stage dementia can no longer contribute to the world in any meaningful way. ^†^	4.61 (1.76)	0.53	0.85	0.20	0.42	0.20	1.20	1.28
7. All residents with dementia should be treated with respect.	6.09 (1.01)	0.40	0.86	0.05	0.53	0.16	0.98	0.93
8. Residents with advanced dementia are no longer true participants in life; instead, they watch from the sidelines. ^†^	5.38 (1.57)	0.50	0.85	0.21	0.73	−0.03	1.31	1.26
16. Residents with dementia who whine a lot should be isolated. ^†^	4.09 (1.57)	0.33	0.86	−0.01	0.41	0.09	1.18	1.23
17. The needs of residents who still have awareness of their environment should take priority over the needs of those who have less awareness. ^†^	4.12 (1.55)	0.34	0.86	0.01	0.51	0.00	1.13	1.19
18. Residents with advanced dementia are no longer persons like you and me, because they do not think and reason logically. ^†^	6.10 (1.24)	0.45	0.85	−0.09	0.81	0.11	1.44	1.25
Factor 3. Psychosocial engagement (Cronbach’s alpha = 0.82, Omega = 0.86)								
9. It is possible for residents with dementia to connect with each other in meaningful ways.	4.60 (1.46)	0.49	0.85	0.41	0.00	0.44	0.86	0.85
10. Residents with dementia want to socialize with the people around them.	4.89 (1.34)	0.62	0.85	0.25	−0.01	0.67	0.64	0.63
11. Residents with dementia can continue to play an important role in their families.	4.11 (1.63)	0.51	0.85	0.23	−0.10	0.61	0.97	0.97
12. Some residents with dementia have had an important role in my life.	4.75 (1.53)	0.41	0.86	0.07	0.04	0.48	1.09	1.11
13. Providing stimulation such as music is very helpful for a resident with end-stage dementia.	5.41 (1.23)	0.51	0.85	−0.02	0.20	0.54	0.79	0.74
14. As dementia advances, residents with dementia no longer experience basic feelings such as pleasure. ^†^	4.57 (1.77)	0.41	0.86	−0.01	0.26	0.31	1.38	1.43
15. Residents with end-stage dementia have some awareness of what is happening around them.	4.02 (1.56)	0.44	0.85	0.16	-0.18	0.61	1.00	0.99
19. Residents with dementia have feelings about their experiences.	5.28 (1.24)	0.49	0.85	−0.26	0.03	0.86	0.79	0.75
20. Most residents with dementia feel the same range of emotions as I do.	5.21 (1.42)	0.54	0.85	−0.22	0.00	0.91	0.92	0.88
Mean (SD) of total score = 95.79 (15.30)								
Total Cronbach’s alpha = 0.86								
Person separation index = 2.35								

^†^ Reverse-coded item; SD: standard deviation; MnSq: mean square.

**Table 3 ijerph-16-04834-t003:** Known-group comparisons of the KPDQ between high-job satisfaction group and low-job satisfaction group (*N* = 269).

Group	KPDQ		
	*n*	Mean (SD)	t (*p*)
High-job satisfaction group	229	96.96 (15.27)	−3.04 (0.003)
Low-job satisfaction group	40	89.10 (13.87)	

KPDQ: Korean version of Personhood in Dementia Questionnaire.

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
