# Peer review of "Psychometric Evaluation of the Korean Version of the Personhood in Dementia Questionnaire Using Rasch Analysis"

_ijerph, 2019, doi:10.3390/ijerph16234834_

Round 1

Reviewer 1 Report

Overall, this paper is a useful addition to the literature. The translation was conducted thoroughly, the sample was decent, and adequate psychometric analyses were conducted. However, I was a bit puzzled why the Rasch analysis was only used for a very specific aspect. See more detailed comments below:

Abstract: Please re-word the first sentence. It is a bit of an exaggeration. Yes, certainly, there would have been a lot of discrimination and abuse against people with dementia, but it sounds here like there has been a unique insight that people with dementia are people who deserve respect. In all societies, elderly have usually been seen as people who one should show respect to. So, perhaps re-word to “There is increasing awareness on the importance to promote behaviors that are consistent with the understanding that individuals with dementia are deserving of adequate respect.” Or something similar along these lines. Abstract: re-word “based on the World Health Organization’s guidelines”. It is written in a way that assumes that the reader is familiar with this or as if it has been mentioned previously. Perhaps re-word to “based on commonly used guidelines by the World Health Organization”. Wording in Abstract, Line 20: the participants were not analyzed, but the data obtained from these participants. Wording in Abstract, Line 21: Rasch does more than test internal consistency reliability. There are more outputs rather than only PSI. Since Rasch assumes unidimensionality, the PCA should be conducted first and therefore mentioned first. Introduction, Line 82: What does BNS mean? Introduction, Line 86: There is no need to mention “(cognitive interviewing)” at the end of the sentence again. Introduction, Line 90: Re-word to “from a parent study that examined…” Introduction, Line 92: The word data is plural, so it should be “Data were collected” Method: Mention the software packages used for the various statistical analyses. Why was the Rasch analysis only conducted to generate PSI? Rasch analysis has the potential to provide much detailed information about item performance and differential item functioning. It is also a bit puzzling that no items had to be deleted. Given the length of the questionnaire and the fact that it was translated across two very different languages, one would typically expect that at least a few items would have to be modified or deleted.

Author Response

Point 1: Abstract: Please re-word the first sentence. It is a bit of an exaggeration. Yes, certainly, there would have been a lot of discrimination and abuse against people with dementia, but it sounds here like there has been a unique insight that people with dementia are people who deserve respect. In all societies, elderly have usually been seen as people who one should show respect to. So, perhaps re-word to “There is increasing awareness on the importance to promote behaviors that are consistent with the understanding that individuals with dementia are deserving of adequate respect.” Or something similar along these lines.

-> Response 1: Thank you for your good suggestion. We revised the first sentence according to your suggestion. (“There is an increasing awareness of the need to promote behaviors consistent with the understanding that individuals with dementia deserve adequate respect”) (p.1).

Point 2: Abstract: re-word “based on the World Health Organization’s guidelines”. It is written in a way that assumes that the reader is familiar with this or as if it has been mentioned previously. Perhaps re-word to “based on commonly used guidelines by the World Health Organization”.

-> Response 2: We revised the phrase according to your suggestion (p.1).

Point 3: Wording in Abstract, Line 20: the participants were not analyzed, but the data obtained from these participants.

-> Response 3: We revised the sentence according to your suggestion. (“The data obtained from a total of 269 participants in 13 long-term care facilities were analyzed") (p.1).

Point 4: Wording in Abstract, Line 21: Rasch does more than test internal consistency reliability. There are more outputs rather than only PSI. Since Rasch assumes unidimensionality, the PCA should be conducted first and therefore mentioned first.

-> Response 4: As you suggested, we added differential item functioning (DIF) and mentioned PSI later (p.1).

Point 5: Introduction, Line 82: What does BNS mean?

-> Response 5: We added full-term of BSN in the sentence. For your information, a BSN degree means a bachelor of science in nursing which is typically acquired through a four-year program at a college or university of nursing (p.2).

Point 6: Introduction, Line 86: There is no need to mention “(cognitive interviewing)” at the end of the sentence again.

-> Response 6: As you suggested, we have excluded the phrase (p.2).

Point 7: Introduction, Line 90: Re-word to “from a parent study that examined…”

-> Response 7: We have changed the phrase according to your suggestion (p.2).

Point 8:  Introduction, Line 92: The word data is plural, so it should be “Data were collected”

-> Response 8: We changed the word from “was” to “were” (p.2).

Point 9: Method: Mention the software packages used for the various statistical analyses. Why was the Rasch analysis only conducted to generate PSI? Rasch analysis has the potential to provide much detailed information about item performance and differential item functioning.

-> Response 9: At the end of the Statistical analyses section, we presented the statistic software packages. Also, we added information about differential item functioning (DIF) (p.4).

Point 10: It is also a bit puzzling that no items had to be deleted. Given the length of the questionnaire and the fact that it was translated across two very different languages, one would typically expect that at least a few items would have to be modified or deleted.

-> Response 10: In the exploratory factor analysis, all item except one were loaded (> 0.40) in at least one factor. The loading value of item 14 (0.31) was lower than desirable criteria, but higher than the minimum recommended item factor loading value of 0.30 (Hair et al. 1998). Also, the results of the confirmatory factor analysis indicated that the three identified factors were appropriate with a good fit. Thus, the research team decided to maintain item 14 (p.4-6).

Reviewer 2 Report

First of all, I congratulate the authors of the manuscript for their great work and effort. It seems important to me that this type of research is published. Next, my recommendations for the authors.

Introduction.

Due to the nature of the study, it seems to me important and pertinent that the authors offer more details about the psychometric properties of the instrument, both in its original validation and in other countries. This would allow the reader to have a better understanding of the adaptation process that the authors performed, as well as to compare the results of the present study in other cultural adaptations or other validations of the instrument. Even, it seems necessary to me that in the literature review they present other instruments that measure the same variable or similar variables. This would help us understand why the authors decided to adapt the PDQ to the Korean context and not another instrument.

Materials and Methods.

I recommend that at the beginning of the Materials and Methods section describe the type of research design used. The characteristic data of the participants (found in section 3.1) should go in section 2.2. I recommend relocating this place section. Participants (section 2.2). The sentence that says “As the number of 101 KPDQ items is 20, the number of study participants should have been more than 120” requires a reference that validates this justification. Many authors indicate that a ratio of 5 participants per item is not enough. Statistical analyses. There are dozens of papers in the psychometric literature that have shown that the most popular measure of internal consistency reliability, Cronbach's alpha coefficient, is seriously flawed. For this reason, I recommend that you add an additional reliability index; preferably the Omega (McDonald, 1999).

Suggested Literature:

Dunn, T., Baguley, T., & Brunsden, V. (2013, in press). From alpha to omega: A practical solution to the pervasive problem of internal consistency estimation. British Journal of Psychology.

McDonald, R. P. (1999). Test theory: A unified approach. Mahwah, NJ: Lawrence Erlbaum Associates.

Results.

4.1. Unidimensionality. It seems necessary to me that the authors present a Table that includes all the items with their factor loads. In addition, I have serious doubts about the method used to calculate the unidimensionality of the instrument. It does not seem to me that the approach of the authors was to explore the possible dimensions of the instrument in the Korean context (analysis carried out with an exploratory factor analysis EFA). It seems to me that the authors wanted to test the unidimensionality of the scale in advance. If that was the intention, it is appropriate to perform a confirmatory factor analysis (CFA). I suggest that the authors perform a confirmatory factor analysis and present their goodness indices, instead of presenting the principal component analysis of residuals.

Results.

This section should be reviewed once the authors make the new analyzes.

Author Response

Point 1: Due to the nature of the study, it seems to me important and pertinent that the authors offer more details about the psychometric properties of the instrument, both in its original validation and in other countries. This would allow the reader to have a better understanding of the adaptation process that the authors performed, as well as to compare the results of the present study in other cultural adaptations or other validations of the instrument. Even, it seems necessary to me that in the literature review they present other instruments that measure the same variable or similar variables. This would help us understand why the authors decided to adapt the PDQ to the Korean context and not another instrument.

-> Response 1: According to your suggestion, we included more persuasive description why we selected PDQ to measure the attitudes toward the persons with dementia in the introduction (p.2). Moreover, additional detailed psychometric properties of the original instrument have been presented at the section of 2.4.1 Instrument (p.3). In other countries except Canada, where the original instrument was developed, the psychometric properties have not yet been examined. We have added a description of another instrument that measures the person-centered attitudes at the Introduction (p.2).

Point 2: I recommend that at the beginning of the Materials and Methods section describe the type of research design used. The characteristic data of the participants (found in section 3.1) should go in section 2.2. I recommend relocating this place section. Participants (section 2.2).

-> Response 2: We added a section (2.1. study design) at the beginning of the Materials and Methods section (p.2). In terms of the characteristic data of the participants, as they are characteristics about demographic information, person-centered climate and job satisfaction of the data analyzed in this study, we decided to maintain those information in section 3.1 (p.5).

Point 3: The sentence that says “As the number of 101 KPDQ items is 20, the number of study participants should have been more than 120” requires a reference that validates this justification. Many authors indicate that a ratio of 5 participants per item is not enough.

-> Response 3: We added a reference for sample size requirement (p.3).

Point 4: Statistical analyses. There are dozens of papers in the psychometric literature that have shown that the most popular measure of internal consistency reliability, Cronbach's alpha coefficient, is seriously flawed. For this reason, I recommend that you add an additional reliability index; preferably the Omega (McDonald, 1999).

-> Response 4: Thank you for the great suggestion. We added omega coefficient in the statistical analysis section and results as you suggested (p.4 & 6).

Point 5: Results. 4.1. Unidimensionality. It seems necessary to me that the authors present a Table that includes all the items with their factor loads. In addition, I have serious doubts about the method used to calculate the unidimensionality of the instrument. It does not seem to me that the approach of the authors was to explore the possible dimensions of the instrument in the Korean context (analysis carried out with an exploratory factor analysis EFA). It seems to me that the authors wanted to test the unidimensionality of the scale in advance. If that was the intention, it is appropriate to perform a confirmatory factor analysis (CFA). I suggest that the authors perform a confirmatory factor analysis and present their goodness indices, instead of presenting the principal component analysis of residuals.

-> Response 5: As you suggested, we conducted an exploratory factor analysis to explore the possible dimensions of the KPDQ and identified 3-factor model. Also, we performed a confirmatory factor analysis and presented model fit indices. Originally the factor analysis was not conducted in the original instrument development study (Hunter et al, 2013). During this revision process, we closely communicated with the person who developed this measure so that three factors were labelled in this study (p.4, 7-8).

Factor 1: agency (beliefs about residents' capacity for self-determination) Factor 2: respect for personhood (beliefs about residents' personhood and moral status) Factor 3: psychosocial engagement (beliefs about residents' capacity for psychological and social engagement)

Round 2

Reviewer 1 Report

Thank you for the thorough revisions.

Reviewer 2 Report

I thank the authors for accepting my recommendations. I congratulate you. I understand that after the changes made, the manuscript is ready to be published.